# High Nitric Oxide Concentration Inhibits Photosynthetic Pigment Biosynthesis by Promoting the Degradation of Transcription Factor HY5 in Tomato

**DOI:** 10.3390/ijms23116027

**Published:** 2022-05-27

**Authors:** Lingyu Wang, Rui Lin, Jin Xu, Jianing Song, Shujun Shao, Jingquan Yu, Yanhong Zhou

**Affiliations:** 1Department of Horticulture, Zijingang Campus, Zhejiang University, 866 Yuhangtang Road, Hangzhou 310058, China; 11716054@zju.edu.cn (L.W.); ruilin@zju.edu.cn (R.L.); 11816045@zju.edu.cn (J.X.); 11816042@zju.edu.cn (J.S.); 15168205026@163.com (S.S.); jqyu@zju.edu.cn (J.Y.); 2Key Laboratory of Horticultural Plants Growth and Development, Agricultural Ministry of China, Yuhangtang Road 866, Hangzhou 310058, China

**Keywords:** nitric oxide (NO), LONG HYPOCOTYL 5 (HY5), S-nitrosylated glutathione reductase (GSNOR), chlorophyll, carotenoid, photosynthetic pigment, *Solanum lycopersicum*

## Abstract

Photosynthetic pigments in higher plants, including chlorophyll and carotenoid, are crucial for photosynthesis and photoprotection. Previous studies have shown that nitric oxide (NO) plays a dual role in plant photosynthesis. However, how pigment biosynthesis is suppressed by NO remains unclear. In this study, we generated NO-accumulated *gsnor* mutants, applied exogenous NO donors, and used a series of methods, including reverse transcription quantitative PCR, immunoblotting, chromatin immunoprecipitation, electrophoretic mobility shift, dual-luciferase, and NO content assays, to explore the regulation of photosynthetic pigment biosynthesis by NO in tomato. We established that both endogenous and exogenous NO inhibited pigment accumulation and photosynthetic capacities. High levels of NO stimulated the degradation of LONG HYPOCOTYL 5 (HY5) protein and further inactivated the transcription of genes encoding protochlorophyllide oxidoreductase C (PORC) and phytoene synthase 2 (PSY2)—two enzymes that catalyze the rate-limiting steps in chlorophyll and carotenoid biosynthesis. Our findings provide a new insight into the mechanism of NO signaling in modulating HY5-mediated photosynthetic pigment biosynthesis at the transcriptional level in tomato plants.

## 1. Introduction

As a small signaling molecule in plants, nitric oxide (NO) plays a significant role in plant growth, development, and stress responses. Plants can produce NO itself through the reductive pathway and the oxidative pathway via nitrite reductase (NR) and NO synthase (NOS)-like, respectively, and such biosynthesis in cellular organisms always occurs in mitochondria, chloroplasts, and peroxisomes [1,2]. A previous study revealed that NO has been turned into a stimulatory signal for the acquisition of photomorphogenic traits in different plant species, regulating processes such as seed germination, hypocotyl elongation, and seedling greening [3]. Further research has demonstrated that NO signaling participates in light-mediated seedling greening by promoting chlorophyll synthesis and chloroplast-related proteins, such as LCHII and PSIA/B, or by decelerating chlorophyll loss in *Arabidopsis*, soybean, wheat, barley, and tomato [4,5,6,7,8]. Moreover, it was demonstrated that NO plays an important role during chloroplast formation and photosynthetic pigment accumulation through a complex crosstalk among phytohormones and phytochromes [8]. However, NO plays a dual role in plants. For a long time, NO was recognized as a kind of polluted gas and was blamed for its toxic effects on plant growth and photosynthesis [9]. It has been determined in *Chlorella vulgaris* that high levels of NO weaken photosynthetic function by repressing the expression of photosynthesis-related genes, including *psdC*, *psaB*, *chlB*, and *rbcL* [10]. However, the in-depth mechanisms underlying NO inhibition of plant photosynthsis are not yet clear.

The enzyme S-nitrosoglutathione reductase (GSNOR) has been demonstrated as another major element that controls levels of NO and NO-derived molecules under diverse physiological conditions [11,12]. NO can bind to the thiol group of reduced glutathione (GSH) to form S-nitrosoglutathione (GSNO), an intracellular NO reservoir that biologically functions as NO throughout cells and organs. Reversibly, GSNO can be decomposed by GSNOR to generate oxidized glutathione (GSSG) and NH_3_ [13]. GSNOR activities and functions have been studied in *Arabidopsis*, sunflower, tomato, pepper, broccoli, and lettuce [11,14,15,16,17]. Results have shown that GSNOR is involved in plant biological processes, including development, metabolism, and stress responses, by regulating NO homeostasis. In *Arabidopsis*, loss of GSNOR causes raised GSNO levels and leads to abnormal growth conditions, such as loss of apical dominance, decreased seed production, and reduced fertility [14]. The chlorophyll content in the loss-of-function *gsnor1–3* mutant was less than in *Col-0* seedlings, and the mutant showed a yellowish phenotype and altered photosynthetic properties [18]. It was indicated that increased NO levels are connected with decreases in photosynthesis rates and chlorophyll contents. Lately, it was found that suppression of GSNOR increased NO accumulation and decreased photosynthesis and reduced plant growth in tomato plants [19]. It remains unclear whether the function of GSNOR in regulating photosynthesis is closely related to the NO level in plants.

De-etiolation involves the simultaneous formation of chlorophyll, carotenoid, lipids, and proteins to assemble the photosynthetic apparatus [20]. Chlorophyll, one of the photosynthetic pigments, is crucial to plant photosynthesis. Chlorophyll biosynthesis and chloroplast formation supply energy to plants during early seedling development. Protochlorophyllide oxidoreductase (POR), a key regulatory enzyme in the chlorophyll biosynthesis pathway, catalyzes the reduction of protochlorophyllide to chlorophyllide, which is the first step in chlorophyll biosynthesis and is strictly light-dependent [21]. There are three distinct isoforms of POR enzymes in *Arabidopsis*, PORA, PORB, and PORC. Among them, the mRNA of *PORA* and *PORB* mainly accumulates in young seedlings, while the expression of *PORC* is significantly induced by light signaling in matured green tissues [22]. Besides chlorophyll, carotenoid is another essential photosynthetic pigment. Although the carotenoid biosynthesis pathway has been researched in many species during fruit ripening, little attention has been paid to it during vegetative growth. Carotenoid is synthesized in plastids, especially chloroplasts of green tissues, together with chlorophyll. Carotenoid and its derivatives are crucial to plant development, stress responses, photosynthesis, and photoprotection [20,23,24]. Phytoene synthase (PSY) catalyzes geranylgeranyl diphosphate (GGPP) into 15-*cis*-phytoene, which is the key step in carotenoid biosynthesis. The transcription of *PSY* is positively correlated with carotenoid accumulation [25]. Three isoforms of *PSY* genes have been found in the tomato genome, *PSY1*, *PSY2*, and *PSY3*. The expression of *PSY1* is enriched in non-photosynthetic tissues of fruit during ripening, while *PSY2* functions mainly in chloroplast-containing green tissues [26]. Recently, *PSY3* was found to be highly expressed in *Arbuscular mycorrhizal* in order to form strigolactone in roots [27]. It was demonstrated that PYS2 has a more important role than PSY1 in carotenoid biosynthesis. The high enzyme activity of green tissue-specific PSY2 is required for photosynthesis [28]. It seems that the transcription levels of *POR*s and *PSY*s are crucial for the biosynthesis of photosynthetic pigments.

During seedling greening, light signaling is the main trigger for the differentiation of etioplasts into chloroplasts. Transcription factors, such as LONG HYPOCOTYL 5 (HY5), REVEILLE 1 (RVE1), CIRCADIAN CLOCK ASSOCIATED 1 (CCA1), GOLDEN2-LIKE (GLK), and PHYTOCHROME INTERACTING FACTOR 1 (PIF1), have been reported to be involved in light-regulated *POR* expression in *Arabidopsis* [29,30,31,32,33]. It is worth mentioning that HY5, cooperating with PIF1, directly binds to the G-box motif of the *PORC* promoter to regulate *PORC* expression and chlorophyll biosynthesis [34]. As with chlorophyll, the transcriptional control of carotenoid biosynthetic genes plays an important role in the regulation of carotenoid production [25]. PIF1 has been reported to regulate carotenoid accumulation by directly binding to the promoter of *PSY* to inhibit its expression in *Arabidopsis*, while HY5 has been found to accumulate in carotenoid production by binding to the same promoter motif antagonistically towards PIF1 [34,35]. HY5 is also known to positively regulate carotenoid biosynthesis in tomato fruits [36]. The transcription factors RIPENING INHIBITOR 1 (RIN1) and STAY-GREEN protein 1 (SRG1) have been demonstrated to interact with the *PSY1* promoter to regulate carotenoid biosynthesis in tomato fruits during ripening [37,38]. Based on these results, it is obvious that light-related transcriptional regulation plays a major role in controlling the coordination of photosynthetic pigment biosynthesis and photosynthesis. However, there are few studies on the role of NO in such progresses. Therefore, understanding the function of NO in the transcriptional regulation of the genes required for the biosynthesis of the two photosynthetic pigments is important for improving plant photosynthesis.

In this report, we used tomato as a model plant to explore the possible function of NO in photosynthesis. The results showed that high levels of NO inhibited chlorophyll and carotenoid accumulation and weakened photosynthetic abilities in tomato leaves. The adverse effects were dependent on the altered function of light-induced transcription factor HY5 caused by NO. This information unveiled a negative role of NO in photosynthetic pigment biosynthesis and provided new insights into regulation at the transcriptional level.

## 2. Results

### 2.1. Photosynthetic Pigment Biosynthesis Is Inhibited by Both Endogenous and Exogenous NO in Tomato

To investigate the mechanism of NO-regulated pigment biosynthesis, *gsnor* mutants were generated by a CRISPR/Cas9 technique for our experiments. Two homozygous gene editing lines, *gsnor#1* and *gsnor#2*, were isolated (Appendix A). The phenotypes of wild-type (WT) and transgenic *gsnor#1* and *gsnor#2* tomato plants were respectively analyzed. In comparison with WT, GSNOR enzyme activity was inhibited by nearly 33% in both *gsnor#1* and *gsnor#2* (Appendix A). GSNOR deficiency caused weakened plant growth, as shown by the shorter plant height, lesser stem thickness, and pale green phenotype (Figure 1A and Appendix A). It also resulted in increased accumulation of NO and attenuated photosynthetic capacity, as indicated by lower values for photosynthesis rate (Pn), stomatal conductance (Gs), internal CO_2_ concentration (Ci), and transpiration rate (Tr) (Figure 1B–D). We determined the contents of chlorophyll and carotenoid, two important photosynthetic molecules. Pigment analysis showed that the chlorophyll contents in *gsnor#1* and *gsnor#2* were 83.1% and 73.0% of that in WT. Likewise, carotenoid contents in *gsnor#1* and *gsnor#2* were 77.6% and 62.4% of that in WT, respectively. (Figure 1E). We identified three distinct POR isoforms coding *Solanum lycopersicum* PORA, PORB, and PORC. Among the three PORs, only the expression of *PORC* was strictly induced by light signaling (Appendix A). We also detected the expression of two paralogs of *PSY*, *PSY1* and *PSY2*, in different tissues. Results showed that *PSY2* was present in greater abundance in leaves than *PSY1* (Appendix A). The expression of *PORC* decreased to 53.8% and 56.1% in *gsnor#1* and *gsnor#2*, respectively, compared to WT plants. Similarly, the expression of *PSY2* reduced to 48.2% and 75.9% in the two transgenic lines (Figure 1F). 

To determine whether decreased pigment content was associated with alteration in NO homeostasis, WT seedlings were treated with various concentrations (10, 100, 200, 500, and 1000 μM) of sodium nitroprusside (SNP, a NO donor) and S-nitroso glutathione (GSNO) as foliar spray. Water was used as a control. The results showed that foliar application of 500 μM SNP and 500 μM GSNO significantly increased the release of intracellular NO, accompanied with the lowest chlorophyll and carotenoid contents (Figure 2A,B and Appendix A). Similar to lack of *GSNOR*, leaves with SNP and GSNO application showed greater fluorescence intensity and weakened photosynthetic capacities, including Pn, Ci, Tr, and Gs (Figure 2C,D and Appendix A). Compared with the control, exogenous NO inhibited photosynthetic pigment production. Approximately 89% of chlorophyll and 88% of carotenoid were detected in the plants treated with 500 μM SNP and 500 μM GSNO (Figure 2E). Simultaneously, the expressions of *PORC* and *PSY2* in SNP- and GSNO-treated plants decreased by over 30% relative to the control (Figure 2F).

All these biochemical observations along with genetic evidence confirm that *GSNOR* modulates intracellular NO accumulation. Both endogenous NO and exogenous NO donors regulate photosynthetic pigment biosynthesis and photosynthesis in tomato plants.

### 2.2. HY5 Mediates NO-Regulated Photosynthetic Pigment biosynthesis 

HY5 is an important light signaling transcription factor. We generated an *hy5* mutant with a CRISPR/Cas9 system (Appendix A). We assessed the phenotype and photosynthetic capacities of the *hy5* mutant as compared with WT plants and found decreased plant height, etiolated leaves, and lower values for Pn, Ci, Tr, and Gs in *hy5* mutant plants (Figure 3A,B and Appendix A). Moreover, there were significant reductions in chlorophyll and carotenoid accumulation in *hy5* (Figure 3C), and the relative expressions of *PORC* and *PSY2* in *hy5* decreased to 70.1% and 61.2% of those in WT (Figure 3D). It was demonstrated in previous studies that HY5 is a crucial and positive regulator in plant photosynthesis. Thus, we determined HY5 protein content and gene expression in the WT and *gsnor#1* and *gsnor#2* transgenic lines. Results showed that knockout of *GSNOR* caused inhibition of HY5 in both transcription and translation levels. The accumulation of HY5 protein and *HY5* gene expression were greatly reduced in *gsnor#1* and *gsnor#2* (Figure 3E,F). To clarify whether the alternation in HY5 was related to GSNOR-modulated changes in NO level, we examined HY5 protein content and gene expression in SNP- and GSNO-treated plants. Similarly, decreased protein accumulation and mRNA levels in HY5 were detected after treatment with exogenous NO donors (Figure 3G,H). Therefore, we conclude that the regulation of HY5 in tomato photosynthetic pigment biosynthesis can be influenced by both *GSNOR* and NO at translation and transcription levels.

### 2.3. NO Accelerates the Degradation of HY5

We next conducted biochemical experiments to test how NO influences HY5 protein levels. A cell-free degradation assay showed that purified recombinant His-HY5 protein incubated with total protein extracts of *gsnor#1* or *gsnor#2* was rapidly degraded in nearly 10 min. However, the same amount of purified recombinant His-HY5 protein was gradually degraded after 60 min incubation in WT plant extracts. Moreover, MG132, a specific inhibitor of the 26S proteasome, prevented the degradation of His-HY5 in the total protein extracts in all cases (Figure 4A). To examine whether His-HY5 degradation is mediated by NO content in plants, foliar applications of 500 μM SNP and 500 μM GSNO were made. Similarly, results showed that the degradation rate of purified recombinant His-HY5 protein accelerated obviously in the presence of NO donors. Specifically, His-HY5 degradation was complete in almost 60 min in the environment of total protein extracts of WT plants treated with H_2_O, while the same amount of His-HY5 was degraded in 30 min and 20 min when incubated with extracts of SNP- and GSNO-treated plants, respectively (Figure 4B). These results indicate that NO mediated the proteolysis of HY5 by reducing HY5 protein stability.

### 2.4. NO Inhibits the Transcriptional Regulation Ability of HY5

As a transcription factor, HY5 activates the expression of *PORC* and *PSY2* by directly binding to G-boxes in their promoters (Appendix A). To explore the role NO played in pigment biosynthesis by affecting HY5 protein stability in tomato plants, we examined whether the transcriptional capacity of HY5 would be regulated by NO signaling. EMSA assays revealed that the ability of HY5 to bind to the sequences containing G-box motifs in the *PORC* and *PSY2* promoters was decreased when purified recombinant His-HY5 protein was incubated with 250 μM GSNO. However, such inhibitory effects of the probe–protein complex caused by GSNO were abolished in the presence of dithiothreitol (DTT) (Figure 5A,B). Obviously, both GSNO and DTT did not work when mutated oligonucleotides were used. We then performed a dual-LUC assay to determine whether the transactivation of *PORC* and *PSY2* by HY5 would be regulated by NO signaling. The results indicated that the significant activation effect of HY5 on the promoters of *PORC* and *PSY2* was reversed both by foliar application of 500 μM SNP and 500 μM GSNO (Figure 5C,D). Taken together, in vivo and in vitro assays all demonstrate that NO signaling reduces *PORC* and *PSY2* gene expression by inhibiting the transactivation ability of HY5.

### 2.5. HY5 Plays a Dominant Role Downstream of NO Signaling in NO-Regulated Pigment Biosynthesis

To confirm that HY5 is essential in the NO-regulated pigment biosynthesis pathway, we silenced *GSNOR* in WT and the *hy5* mutant, using the virus-induced gene silencing (VIGS) technique, which decreased transcript levels of *GSNOR* by 75–85% (Appendix A). We found that the *hy5* mutant was almost insensitive to the inhibition of pigment biosynthesis by NO. Compared to pTRV plants, silencing of *GSNOR* (pTRV-*GSNOR*) showed reduced chlorophyll and carotenoid contents, as in *gsnor* mutant plants. However, silencing of *GSNOR* did not decrease the accumulation of chlorophyll and carotenoid in *hy5* plants (Figure 6A,B). Next, the SNP and GSNO treatments were administered as spray applications to determine whether NO had an effect when *GSNOR* was silenced. The results showed that, compared to the H_2_O treatment, there were no significant reductions in chlorophyll and carotenoid contents in *hy5* mutant plants, regardless of spraying with 500 μM SNP or 500 μM GSNO (Figure 6C,D). In addition, the treatment of exogenous NO donors reduced the expression of *PORC* and *PSY2* in WT compared to the control, while in *hy5*, the transcripts of *PORC* and *PSY2* showed no changes regardless of whether SNP or GSNO was used (Appendix A). These results suggest that HY5 is a key factor in and plays a dominant role downstream of NO signaling in the regulation of pigment biosynthesis by NO in tomato.

## 3. Discussion

The function of NO in the regulation of plant growth and development has been reported previously. However, the regulatory mechanism that realizes the role of NO in regulating plant photosynthesis still needs to be investigated. In this study, we found that, as the transcriptional activator of *PORC* and *PSY2*, HY5 was involved in the biosynthesis of chlorophyll and carotenoid in tomato plants. We further demonstrated that excess NO could stimulate the degradation of HY5 protein and further decelerate pigment accumulation and photosynthesis (Figure 7). 

As many molecular targets of NO have been researched in plants, how plants maintain a steady-state concentration of intracellular NO is an important question. NO homeostasis in plants depends on a delicate balance between its production and scavenging. GSNOR is one of the pathways involved in the regulation of NO homeostasis [39]. GSNOR can directly act on GSNO to reduce NO levels, which is extremely important to maintain NO homeostasis. Additionally, phytohormone signaling might be involved in GSNOR-mediated plant growth and development. The ethylene productivity rate was increased in *GSNOR*-RNAi plants [19]. It was reported that ethylene facilitated NO generation by activation of both NR and NOS-like in *Arabidopsis* and *Tagetes erecta* L. [40,41]. The crosstalk between NO and other phytohormones might be a potential mechanism that maintains the high NO levels in *GSNOR*-knockdown plants. The *GSNOR*-knockdown mutants used in this study were disabled to scavenge superfluous intracellular NO and maintained a high level of NO (Figure 1B,C). A pale green phenotype was found in *gsnor#1* and *gsnor#2*, and further pigment assays demonstrated that chlorophyll and carotenoid contents in *gsnor* mutants were lower than in WT (Figure 1A,E). Based on these observations, we suggested that the alternation in pigment contents was regulated by intracellular NO. As with other free radical molecules, NO has beneficial or detrimental influences depending on its intracellular concentration. A study on tomato plants showed that a treatment of 1 M SNP inhibited plant growth, which could have been a consequence of a supraoptimal concentration of SNP [42], while 200 μM SNP and 1 M GSNO showed a negative effect on chlorophyll fluorescence parameters [43]. In wheat, 100 μM SNP-treated seedlings contained more chlorophyll than the controls [3]. In general, the concentration of SNP used in previous studies inducing chlorophyll accumulation was lower than 100 μM. In our study, different concentration gradients of NO donors were examined. The results showed that a low concentration (lower than 100 μM) of SNP or GSNO causes the accumulation of chlorophyll and carotenoid, while a high concentration (higher than 100 μM) of SNP or GSNO inhibits accumulation. Among these concentrations, 500 μM of SNP or GSNO induced the most significant decrease in pigment content (Appendix A). Meanwhile, a massive accumulation of NO was determined in 500 μM SNP- or GSNO-treated plants (Figure 2A–C). Furthermore, the reduction in pigment biosynthesis was related to suppressed *PORC* and *PSY2* gene expression by excess NO (Figure 1F and Figure 2F). The finding that NO regulates the expression of genes involved in photosynthetic pigment biosynthesis is a breakthrough in this research.

HY5 is the main regulator of many developmental processes, including chloroplast development, pigment accumulation, nutrient assimilation, and stress responses. In our early studies, it was demonstrated that HY5 plays an important role in plant growth, cold tolerance, photoprotective response, starch degradation, and mycorrhizal symbiosis in tomato [44,45,46,47]. The data presented here show that HY5 is the key factor that regulates chlorophyll and carotenoid accumulation in tomato leaves. The results of in vivo and in vitro assays demonstrated that HY5 could bind to G-box motifs in *PORC* and *PSY2* promoters and positively regulate their transcription (Appendix A). Chlorophyll and carotenoid contents were higher in WT than in *hy5* (Figure 3C). The existence of HY5 also promoted the photosynthetic capacity of WT as compared to *hy5* (Figure 3B and Appendix A). 

In addition, there are several external regulatory signals which influence plants through the HY5-mediated pathway. High temperature repressed anthocyanin biosynthesis by inducing the degradation of HY5 protein [48]. Ethylene suppressed HY5 accumulation and stability via 26S proteasome-mediated degradation to promote hypocotyl growth [49]. Strigolactone promoted *HY5* expression and HY5 protein accumulation to inhibit hypocotyl elongation [50]. We found that the transcript levels of *HY5*, along with HY5 protein levels, were lower in NO-accumulated *gsnor#1* and *gsnor#2* plants than in WT (Figure 3E,F). The treatment with SNP and GSNO also caused a reduction in HY5 transcriptional and protein level (Figure 3G,H). Further cell-free degradation assays confirmed that both endogenous and exogenous NO weakened HY5 protein stability (Figure 4). In *Arabidopsis*, it was reported that UV-B induced the activation of *HY5* expression and the accumulation of HY5 protein, strengthened interactions between HY5 and *CHS* promoters, and finally increased anthocyanin production [51]. In addition, cryptochrome and cytokinin signaling can increase the accumulation of HY5 at the protein level to activate the transcription of anthocyanin biosynthetic enzymes [52]. In this research, we revealed that NO decreased the accumulation of chlorophyll and carotenoid via inhibiting the HY5-mediated transcriptional activation of the pigment synthesis genes *PORC* and *PSY2* (Figure 1F, Figure 2F, and Figure 5). It is worth mentioning that the pigment concentration in *hy5* did not show a significant decline compared to that in WT in the presence of NO. Additionally, there seemed to be a minor decrease in the pigment concentration of *hy5* once NO was introduced (Figure 6). There might be other factors downstream of NO that regulate pigment biosynthesis while HY5 still plays the dominant role. 

The effects of NO on gene expression are post-translational modifications (PTMs) of transcription factors, regulatory proteins, or nuclear proteins through S-nitrosylation, tyrosine nitration, and metal nitrosylation [53]. A microarray analysis showed that 10% of genes that respond to SNP treatment are transcription factors [54]. The potential S-nitrosylation and tyrosine nitration sites of HY5 were predicted through the application of GPS-SNO 1.0 (http://sno.biocuckoo.org/ (accessed on 27 October 2020)) and GPS-YNO2 (http://yno2.biocuckoo.org/ (accessed on 22 April 2021)). Results showed that there was no cysteine thiol that can be targeted by NO in the amino acid sequence of HY5, while HY5 protein was predicted possibly to be nitrated at Tyr-110. It is meaningful to explore the potential PTM of HY5 caused by NO and confirm the alternation of HY5 transcriptional activity.

In summary, our data demonstrated that accumulated intracellular NO works as an inhibitor of plant photosynthetic functions during vegetative growth by inactivating the expression and translation of HY5. The transcripts of *PORC* and *PSY2* were suppressed concomitant with the degradation of HY5. These findings provide a new insight into the mechanism for NO signaling in modulating HY5-mediated photosynthetic pigment biosynthesis at the transcriptional level in tomato plants. Further work on the potential NO-induced PTMs are needed to ascertain the regulatory mechanism of NO with respect to photosynthesis in tomato plants.

## 4. Materials and Methods

### 4.1. Plant Materials and Growth Conditions

Tomato seeds of *Solanum lycopersicum* L. cv. “Condine Red”, “Ailsa Craig” were used as the wild type in this study. Gene function loss and gain were generated in stable tomato lines through gene-editing and over-expression approaches, respectively. Transgenic plants overexpressing *HY5* (*HY5*-OE) were generated as described by Wang et al. [55]. To generate the *HY5* and *GSNOR* CRISPR/Cas9 vector, primer sequences were designed using the web tool CRISPR-P [56]. The synthesized sequences were annealed and inserted into the AtU6-sgRNA-AtUBQ-Cas9 vector at the *Bbs*I site and the reconstructed AtU6-sgRNA-AtUBQ-Cas9 cassette was inserted into the *Hind*III and *Kpn*I sites of the pCAMBIA1301 binary vector. The resulting confirmed plasmids were transformed into *Agrobacterium tumefaciens* strain GV3101 by electroporation and then introduced into WT plants via a method previously described [57]. After confirmation with PCR and DNA sequencing (Appendix A), two independent homozygous T2 lines of *gsnor* and one of *hy5* mutants were used in this study.

Seedlings were grown in plots filled with a mixture of peat and vermiculite (3:1, *v*/*v*) and watered with Hoagland’s nutrient solution. The growth conditions were as follows: white light (600 μmol m^−2^ s^−1^ photosynthetic photo flux density) applied with a 12 h photoperiod and a temperature of 25 °C/20 °C (day/night). At the 5-leaf stage, the plants were used for the experiments.

### 4.2. Pharmacological Treatments

Tomato seedlings at the 5-leaf stage were treated with 500 μM SNP (Sigma-Aldrich, St. Louis, MO, USA) or 500 μM GSNO (Sigma-Aldrich, St. Louis, MO, USA) as foliar spray once every other day. SNP and GSNO were diluted with double-distilled water. Plants treated with double-distilled water were used as controls.

### 4.3. NO Content Assays and GSNOR Activity Measurements

Intracellular NO levels were detected by confocal microscopy using the NO-specific fluorescent probe, 3-Amino,4-aminomethyl-2′,7′-difluorescein, diacetate (DAF-FM-DA, Beyotime, China), as previously described elsewhere [4]. Briefly, leaf discs (5 mm in diameter) were mixed with 10 μM DAF-FM-DA prepared in 10 mM PBS (pH 7.4) and incubated in the dark at 37 °C for 30 min. After washing off redundant probe with 10 mM PBS three times, fluorescence was observed under a laser scanning confocal microscope (Nikon A1 plus, Japan) at an excitation wavelength of 488 nm and an emission wavelength of 525 nm. Treatments were repeated 8 times. 

NO content was further detected using Griess reagent as described previously with slight modifications [58]. Leaves (0.1 g) were ground in liquid nitrogen and homogenized in 500 μL of 50 mM cool acetic acid buffer (pH 3.6, containing 4% zinc diacetate). The homogenates were centrifuged twice at 10,000× *g* for 10 min at 4 °C. The supernatant was then collected. A mixture of 200 μL of filtrate with 200 μL of Greiss reagent (Sigma-Aldrich, St. Louis, MO, USA) was incubated at room temperature for 30 min. Absorbance was determined at 540 nm. The NO concentration was calculated against a standard curve for NaNO_2_.

GSNOR enzyme activity was determined as described previously [59]. Leaves (0.1 g) were ground in liquid nitrogen and homogenized in extraction buffer (50 mM HEPES, pH 8.0; 20% glycerol; 10 mM MgCl_2_; 1 mM EDTA; 1 mM EGTA; 1 mM benzamidine; and 1 mM ε-aminocaproic acid). The homogenate was centrifuged at 16,000× *g* for 15 min at 4 °C. The supernatant was collected using spin columns (Thermo Fischer Scientific, Waltham, MA, USA). The reaction system included 30 μL protein samples and 300 μL reaction buffer (20 mM Tris-HCl, pH 8.0; 0.2 mM NADH; 0.5 mM EDTA; 400 μM GSNO). The activity of GSNOR was determined by detecting the change in absorbance caused by the decomposition of NADH at 340 nm using a PerkinElmer 2300 EnSpire™ Multilabel Reader. The background in the absence of GSNO was eliminated.

### 4.4. Recombinant Protein and EMSA

The full-length coding region of HY5 was PCR-amplified using the primers listed in Appendix A, then the product was digested with *BamH*I and *Sac*I and cloned into pET-32a vectors. The recombinant His-fusion protein was transformed into *Escherichia coli* BL21 (DE3) and purified following the instructions of the Novagen pET purification system. For EMSA assays, the primers of probes used are listed in Appendix A. Probes were biotin end-labeled according to the instructions of the Biotin 3′ End DNA Labeling Kit (Thermo Fisher Scientific, Waltham, MA, USA) and annealed to a double-stranded DNA probe following the steps described by Jiang et al. [45]. EMSAs of the HY5–DNA complexes were performed using the LightShift Chemiluminescent EMSA Kit (Thermo Fisher Scientific, USA). Briefly, His-HY5 proteins, some of which were treated with 250 μM GSNO and DTT, were incubated together with biotin-labeled probes in reaction mixtures with or without competitors or mutant competitors for 20 min at room temperature and separated on 6% native polyacrylamide gels.

### 4.5. Total RNA Extractions and Gene Expression Analysis

Total RNA of tomato leaves was extracted using a SteadyPure Universal RNA Extraction Kit (AG21017, Accurate Biotechnology, Hunan, China) according to the manufacturer’s instructions. Total RNA (1 μg) was reverse transcribed using an Evo M-MLV RT Kit Mix for qPCR (AG11706). The gene-specific primer pairs are shown in Appendix A. RT-qPCR was performed in a Roche LightCycler 480 real-time PCR system (Roche, Basel, Switzerland) using SYBR Green Premix Pro Taq HS qPCR Kit (AG11701). The RT-qPCR reaction was run at 95 °C for 3 min, followed by 40 cycles of denaturation for 30 s at 95 °C, annealing for 30 s at 58 °C and extension for 1 min at 72 °C. The tomato *Actin* was used as an internal control to calculate the relative transcript levels of target genes [60].

### 4.6. Protein Degradation Assay

The total protein extracts were prepared from WT, *gsnor#1*, *gsnor#2*, and foliar spray-treated seedlings. Leaf samples (0.3 g) were ground in liquid nitrogen and resuspended in extraction buffer (25 mM Tris, pH 7.5; 10 mM MgCl_2_; 5 mM DTT; 10 mM NaCl; and 10 mM ATP) modified according to Lin et al. [61]. Equal amounts of His-HY5 (about 200 ng) were incubated in 200 μL of total protein extracts (about 20 μg) at room temperature and reactions were stopped by adding an equal volume of 2× protein gel-loading buffer. MG132 was added as indicated. The samples were taken at the indicated intervals.

### 4.7. Protein Extraction and Immunoblot Analysis

Immunoblot analysis was performed as described previously [47]. For protein extraction, leaf samples were ground into powder in liquid nitrogen and homogenized in extraction buffer (100 mM HEPES, pH 7.5; 5 mM EDTA; 5 mM EGTA; 10 mM Na_3_VO_4_; 10 mM NaF; 50 mM β-glycerophosphate; 1 mM phenylmethylsulphonyl fluoride; 10% (*v*/*v*) glycerol; 7.5% (*w*/*v*) polyvinylpolypyrrolidone; and 0.2% (*v*/*v*) β-mercaptoethanol), followed by centrifugation at 12,000× *g* for 20 min. Proteins adjusted to equal concentration were then mixed with 4× loading buffer and denaturized at 95 °C for 10 min. SDS–polyacrylamide gel (12% [*w*/*v*]) electrophoresis (SDS-PAGE) was performed to separate the protein extracts. For HY5 detection, we used an HY5-specific antibody (Shanghai Jiayuan Bio Co., Shanghai, China), which was detected using the ECL western detection kit (GE Healthcare, Boston, MA, USA). HSP70 antibody (Beijing Protein Innovation Co., Ltd., Beijing, China) was used as a control.

### 4.8. Pigment Analysis

Total chlorophyll and carotenoid contents were extracted from 0.3 g leaf discs (10 mm in diameter) with 10 mL 80% acetone in complete darkness over 2 days. Absorbance at 663, 645, and 445.5 nm of the extractions was measured on a PerkinElmer 2300 EnSpire™ Multilabel Reader. The concentrations in the extracts were calculated using MacKinney’s coefficients:C_Chl_(μg mL^−1^) = 20.41 A_645_ − 5.13 A_663_
C_Car_ (μg mL^−1^) = 4.69 A_445_._5_ − 0.268 (A_663_ + A_645_) 
$$\mathrm{chlorophyll}/\mathrm{carotenoid}\;\mathrm{content}\;(\mu g\;g^{-1})=\frac{C\times \;V\;\times \;N}{W}$$
where V = volume of extraction buffer, N = dilution factor, and W = weight of leaf samples.

### 4.9. ChIP Assay

ChIP assays were performed according to the instructions for the EpiQuik Plant ChIP Kit (P-2014; Epigentek, New York, NY, USA). Approximately 1.5 g of leaf samples was harvested from *HY5*-OE and WT plants. Chromatin was immunoprecipitated with an anti-HA antibody (Pierce, Rockford, IL, USA) and goat anti-mouse IgG (Millipore, Burlington, MA, USA) was used as a negative control. ChIP-qPCR was performed with primers for *PORC* and *PSY2* promoters, which are listed in Appendix A.

### 4.10. Yeast One-Hybrid Assay

Fragments of the *PORC* promoter (−1060 to −827) and the *PSY2* promoter (−501 to −126) were amplified and inserted into the pBait-AbAi vector. After digestion with *Bbs*I, the pBait-AbAi vectors were integrated into Y1HGold to generate bait–reporter strains. The positive clones were selected on SD/-Ura medium, and the minimal inhibitory concentration of AbA was confirmed. The HY5 ORF was fused to pGADT7 to achieve a prey construct that was guided into reporter strains, which were then grown on SD/-Leu medium at 30 °C for 72 h. The positive clones were diluted in double-distilled water to OD600 = 0.3. The suspension was spotted on SD/-Leu with or without AbA. The primers used to generate the constructs are listed in Appendix A.

### 4.11. Dual-Luciferase Transactivation Assay

The full length of HY5 was amplified into the vector pGreen II 0029 62-SK and the promoters of *PORC* (−1186 to −882) and *PSY2* (−2089 to −1580) were cloned into the vector pGreen II 0800-LUC. The recombinant plasmids were electroporated into *A. tumefaciens* GV3101: pSoup. Tobacco leaves were infiltrated with Agrobacterium cultures and harvested after 3 days. The activities of firefly luciferase (LUC) and *Renilla* luciferase (REN) were analyzed with dual-luciferase assay reagents (Vazyme Biotech Co., Nanjing, China). The primers used to generate the constructs are listed in Appendix A.

### 4.12. Virus-Induced Gene Silencing

Tobacco rattle virus vectors (pTRV1 and pTRV2) were used for the virus-induced gene silencing (VIGS) of tomato genes and the pTRV-*GSNOR* plasmid was constructed as described in Lv et al. [62]. VIGS was performed by infiltration into the fully expanded cotyledons of WT and *hy5* seedlings with a mixture of *A. tumefaciens* carrying pTRV1 and pTRV2 recombinant plasmid (1:1, *v*/*v*, OD600 = 0.6). RT-qPCR was performed to evaluate the gene silencing efficiency. The primers used to generate the constructs are listed in Appendix A. Empty pTRV2 vectors were used as controls.

### 4.13. Photosynthetic Parameter Analysis

Photosynthetic parameters, including photosynthesis rate (Pn), internal CO_2_ concentration (Ci), transpiration rate (Tr), and stomatal conductance (Gs), were measured using a portable photosynthesis measurement system (LI-6400; LI-COR, Lincoln, NE, USA). The specific parameters were set as follows: 600 μmol m^−2^ s^−1^ photosynthetic photon flux density, 380 μmol mol^−1^ CO_2_ concentration, 23 °C, 85% relative humidity.

## Figures and Tables

**Figure 1 ijms-23-06027-f001:**
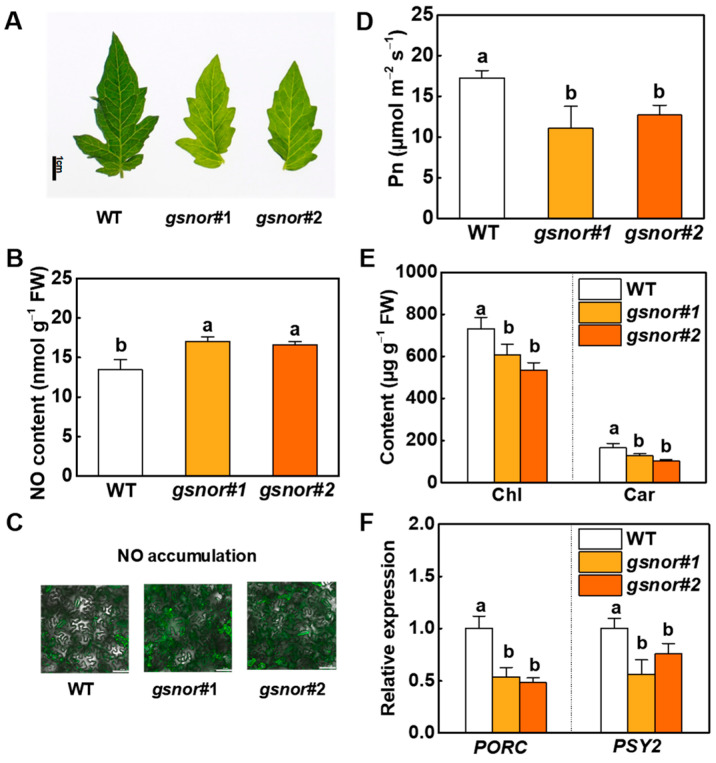
GSNOR regulates photosynthetic pigment biosynthesis through endogenous nitric oxide (NO) in tomato plants. (**A**) Leaf phenotypes of the wild type (WT), *gsnor#1*, and *gsnor#2* plants. Bar represents 1 cm. (**B**,**C**) Nitric oxide (NO) accumulation, (**D**) photosynthesis rate (Pn), (**E**) chlorophyll and carotenoid contents of leaves, and (**F**) relative transcript levels of PROTOCHLOROPHYLLIDE OXIDOREDUCTASE (PORC) and PHYTOENE SYNTHASE (PSY2) in WT, *gsnor#1*, and *gsnor#2*. For (**C**), NO accumulation in leaves was visualized using a NO-specific fluorescent probe, 3-Amino,4-aminomethyl-2′,7′-difluorescein, and diacetate (DAF-FM-DA), and fluorescence was photographed with a laser scanning confocal microscope (Nikon A1plus, Nikon, Tokyo, Japan). Representative images were selected from eight replicates; horizontal bars = 50 μm. For other determinations, data are shown as means ± SD (*n* = 3). Different letters indicate significant differences at *p* < 0.05 according to Tukey’s test.

**Figure 2 ijms-23-06027-f002:**
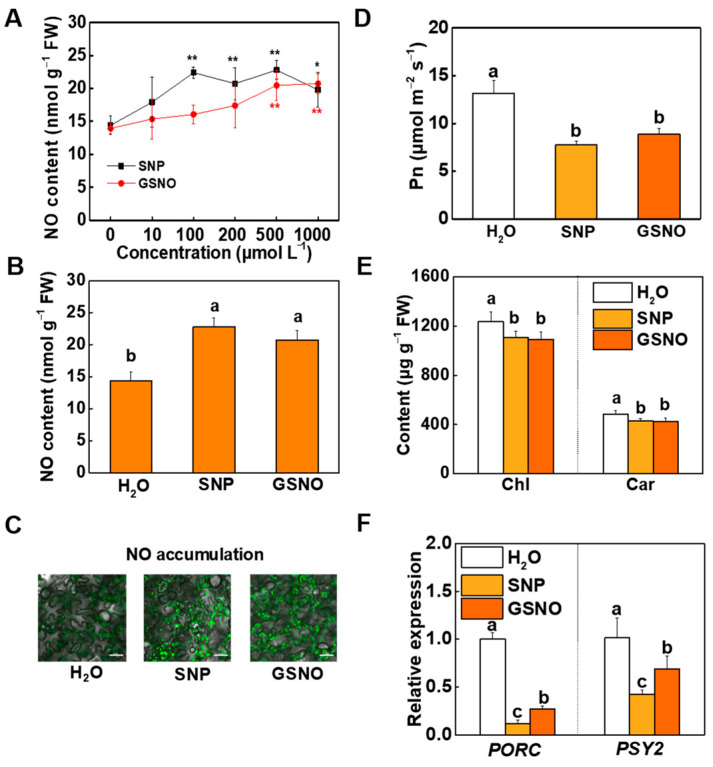
Exogenous NO donors regulate plant photosynthetic pigment biosynthesis. (**A**) NO levels induced by different concentration gradients of sodium nitroprusside (SNP) and S-nitroso glutathione (GSNO). (**B**,**C**) NO accumulation, (**D**) Pn, (**E**) chlorophyll and carotenoid content of leaves, and (F) relative transcript levels of *PORC* and *PSY2* in tomato plants treated with H_2_O, 500 μM SNP, and 500 μM GSNO. NO, chlorophyll and carotenoid content were assayed 5 days after being treated with H_2_O, SNP, and GSNO, while Pn was determined at 3 days after the commencement of exogenous NO treatment. For (**C**), representative images were selected from eight replicates of each treatment; horizontal bars = 20 μm. For other determinations, data are shown as means ± SD (*n* = 3). For (**A**), asterisks indicate statistically significant differences from the control (0 μM SNP or GSNO) with respect to the means (**, *p* < 0.01; *, *p* < 0.05). For (**B**) and (**E**,**F**), different letters indicate significant differences at *p* < 0.05 according to Tukey’s test.

**Figure 3 ijms-23-06027-f003:**
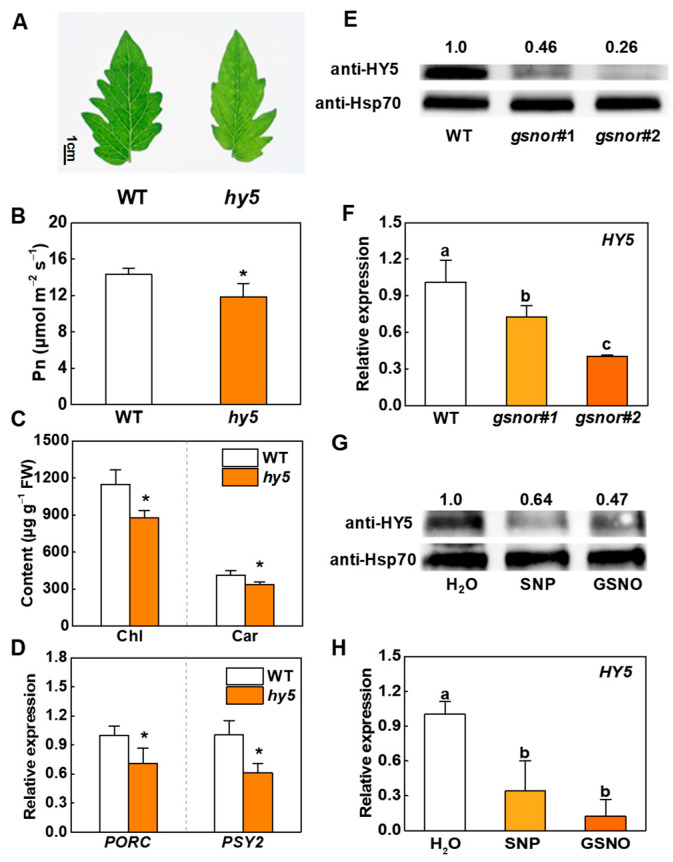
HY5 is required for NO-regulated plant photosynthetic pigment biosynthesis. (**A**) Leaf phenotypes displayed for WT and *hy5*. Bar represents 1 cm. (**B**) Pn, (**C**) chlorophyll and carotenoid content of leaves, and (**D**) relative transcript levels of *PORC* and *PSY2* in WT and *hy5*. (**E**,**F**) HY5 accumulation and relative transcript levels of (ELONGATED HYPOCOTYL5) *HY5* in WT, *gsnor#1*, and *gsnor#2*. (**G**,**H**) The effects of SNP and GSNO on HY5 accumulation and relative transcript levels of *HY5*. Data are shown as means ± SD (*n* = 3). For (**B**–**D**), asterisks indicate statistically significant differences from the control (WT) with respect to the means (*, *p* < 0.05). For (**F**,**H**), different letters indicate significant differences at *p* < 0.05 according to Tukey’s test.

**Figure 4 ijms-23-06027-f004:**
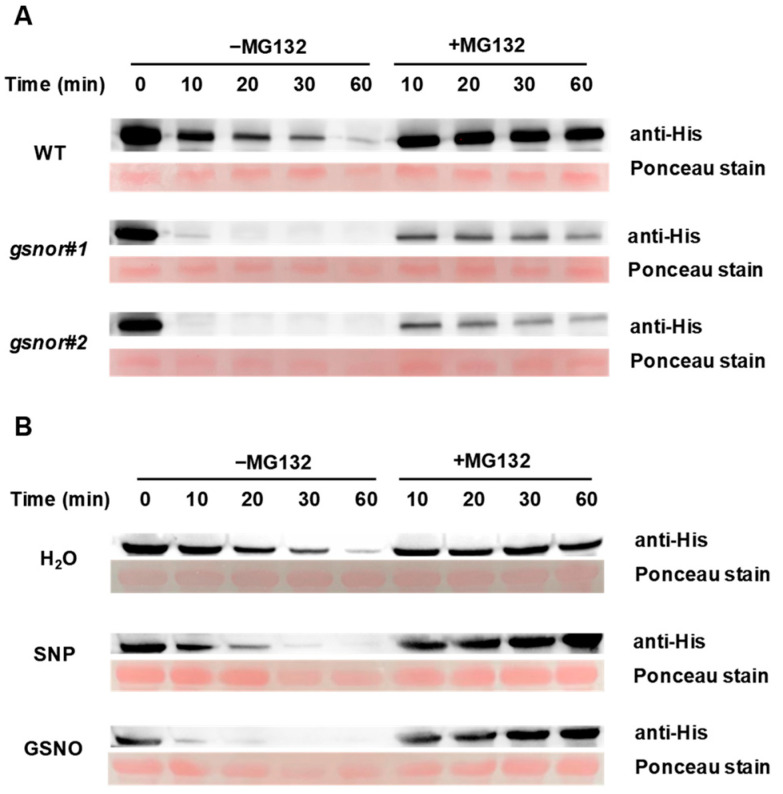
NO signalling promotes HY5 degradation. (**A**) Cell-free degradation assay displayed in WT, *gsnor#1*, and *gsnor#2* leaf extracts. (**B**) Cell-free degradation assay results displayed for H_2_O-, SNP-, and GSNO-treated leaf extracts. The degradation of His-HY5 was blocked in the presence of 50 µM MG132 (+MG132).

**Figure 5 ijms-23-06027-f005:**
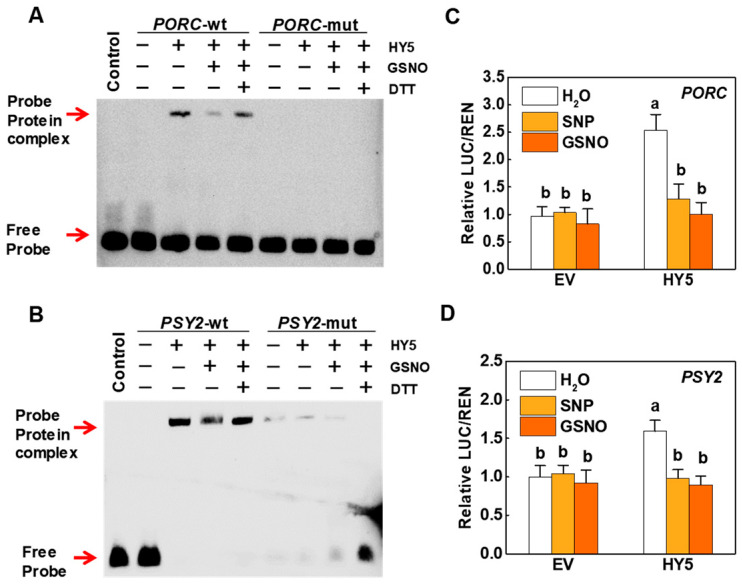
NO suppressed interaction between HY5 and the promoters of *PORC* and *PSY2*. (**A**,**B**) EMSA assay. After 30 min of incubation with or without 250 μM GSNO, the His-HY5 recombinant protein was incubated with biotin-labeled wild-type (wt) or mutant (mut) *PORC* and *PSY2* oligos. The protein purified from the empty vector was used as a negative control. DTT was used as a reducing agent. (**C**,**D**) The regulatory effects of HY5 on the promoters of *PORC* and *PSY2* as determined by dual-luciferase assay. Tobacco (*Nicotiana benthamiana*) leaves were infiltrated, and firefly LUC and *Renilla* LUC were assayed 3 days after infiltration. The LUC/REN ratio of the empty vector (EV) plus promoter treated with H_2_O was set as 1. Data are shown as means ± SD (*n* = 6). Different letters indicate significant differences at *p* < 0.05 according to Tukey’s test.

**Figure 6 ijms-23-06027-f006:**
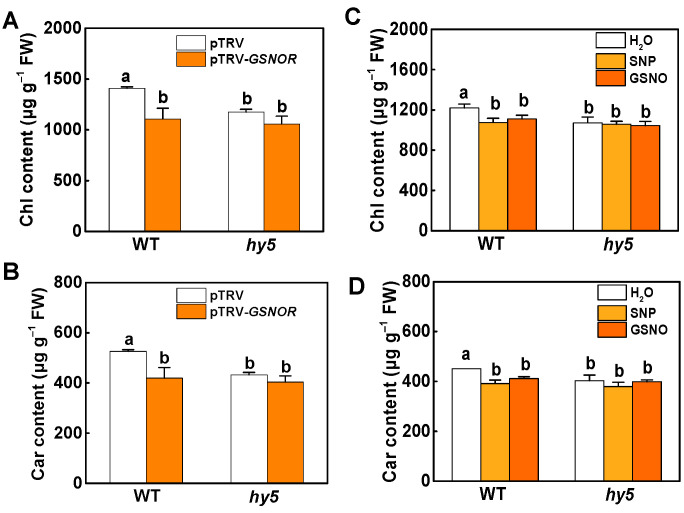
HY5 operates downstream of NO to modulate plant photosynthetic pigment biosynthesis. (**A**,**B**) Chlorophyll and carotenoid content of leaves in WT and *hy5* in which *GSNOR* was silenced by means of the VIGS technique. (**C**,**D**) Chlorophyll and carotenoid content of leaves in WT and *hy5*, 5 days after being treated with H_2_O, SNP, and GSNO. Data are shown as means ± SD (*n* = 3). Different letters indicate significant differences at *p* < 0.05 according to Tukey’s test.

**Figure 7 ijms-23-06027-f007:**
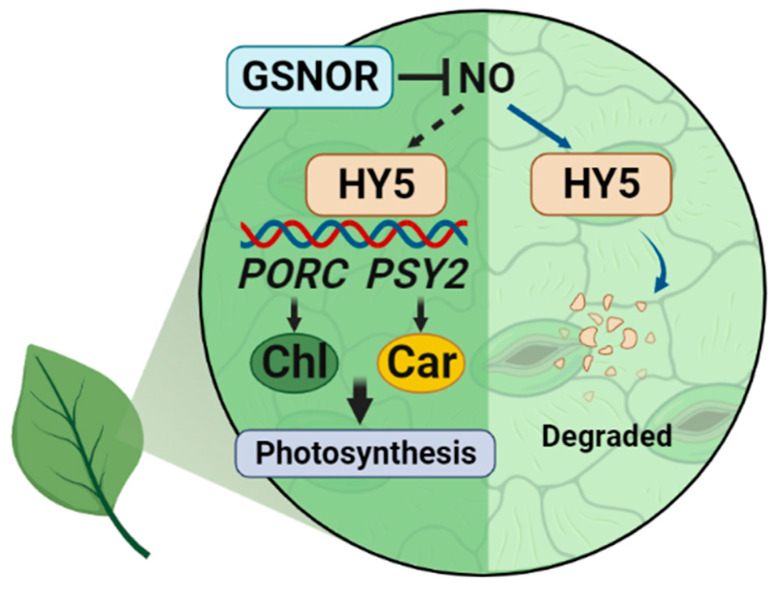
A working model of how NO suppresses HY5-mediated photosynthetic pigment biosynthesis in tomato. Excess NO, which can be eliminated by GSNOR, reduces HY5 stability to inactivate the transcription of pigment biosynthetic genes *PORC* and *PSY2*, thereby suppressing chlorophyll and carotenoid biosynthesis and photosynthesis.

## Data Availability

Not applicable.

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
