# Peer review of "High Nitric Oxide Concentration Inhibits Photosynthetic Pigment Biosynthesis by Promoting the Degradation of Transcription Factor HY5 in Tomato"

_ijms, 2022, doi:10.3390/ijms23116027_

Round 1

Reviewer 1 Report

The manuscript entitled “Nitric oxide inhibits photosynthetic growth by promoting the degradation of transcription factor HY5 in tomato” by Wang et al. provides an unknown function of NO in regulating HY5 transcription factor that transcriptionally activates the genes involved in photosynthetic pigment biosynthesis. While the manuscript elucidates some novel aspects in the NO-HY5 pathway involved in pigment biosynthesis, I have some concerns to be addressed:

Major:

  1. In the 2.5 subsection of the manuscript, the authors mentioned HY5 is downstream of the NO functions. If this hypothesis is true, I believe the authors should observe some phenotype in the presence/accumulation of NO. But in figure 7, the hy5 mutant doesn’t display any significant pigment concentration changes.
  2. To validate the findings of figure 7C and D, I believe authors can show the expression pattern of the PORC and PSY2 genes in hy5 mutant in the presence of SNP and GSNO. Since the treatment of exogenous NO reduces the expressions of PORC and PSY2 in the WT background up to 50~80%, it is essential to see the profile of those genes in the hy5 background in the presence of exogenous NO.
  3. What is the phenotype of the HY5-OE plant? Does it show resistance to NO?
  4. While quantifying the fluorescence values with DAF-FM-DA staining (Figure 1E and F), whether the authors calibrated the autofluorescence due to chlorophyll? In figure 1F, the “horizontal scale bar” is missing.
  5. The expression level of the PORC and PSY2 in the hy5 background is around 50~60% of WT. From this observation, it can be speculated that there are other factors which maintain a stable expression of PORC and PSY2 during pigment biosynthesis. So, in that case, HY5 is not indispensable in the regulation of pigments biosynthesis by NO. At this stage, it is too rudimentary to conclude that HY5 is the only candidate downstream of NO pathway.

Minor:

  1. I wonder whether the authors included the possibility of COP1 in degradation of HY5 in this NO-HY5 mediated pathway that fine-tunes pigment biosynthesis.
  2. English correction and typos need to be corrected: For example, in line #31, “such biosynthesis in cellular always occurs”, I guess it would be cellular organisms.
  3. Line #36, light-mediated seedling greening. The “d” is missing from mediated.
  4. In figure S2B, what are MG, BK, BK+3, BK+7? Please mention it in the legend.
  5. In figure 5, the legend “A” is missing.
  6. In figure 5, please mention which panel is H20, SNP and GSNO treatment. Saying them in the original image file in figure 5B is not enough.

Author Response

Response to Reviewer 1 Comments

Thank you for your careful and patient review of our manuscript. Your positive comments and valuable suggestions are greatly appreciated. In response to your criticisms and suggestions, we have completed several additional experiments and revised the manuscript accordingly. Our responses to your individual criticisms are follows:

Major:

Point 1: In the 2.5 subsection of the manuscript, the authors mentioned HY5 is downstream of the NO functions. If this hypothesis is true, I believe the authors should observe some phenotype in the presence/accumulation of NO. But in figure 7, the hy5 mutant doesn’t display any significant pigment concentration changes. 

Response 1: Thank you for making this critical point. As you mentioned in major point 5, it is probably that HY5 is not indispensable in the regulation of pigments biosynthesis by NO. In response to your concern, we have checked our original data and made a little modification in figure 6C (figure 7C in the original version). We found that there seemed to be an insignificant pigment concentration decline in hy5 mutant in the presence/accumulation of NO. There should be other factors, whose function are minor, that response to NO signaling in modulating pigment biosynthesis. The 2.5 subsection has been improved in the revised MS.

Point 2: To validate the findings of figure 7C and D, I believe authors can show the expression pattern of the PORC and PSY2 genes in hy5 mutant in the presence of SNP and GSNO. Since the treatment of exogenous NO reduces the expressions of PORC and PSY2 in the WT background up to 50~80%, it is essential to see the profile of those genes in the hy5 background in the presence of exogenous NO.

Response 2: Thanks for your suggestion. We have carried out an additional experiment to verify the expressions of PORC and PSY2 in both WT and hy5 background in the presence of exogenous NO (Figure S9). The description of the results has been added in the revised article in item 2.5.

Point 3: What is the phenotype of the HY5-OE plant? Does it show resistance to NO?

Response 3: Thanks for your concern. In our early experiments, we have found that HY5-OE plants show resistance to NO. But unfortunately, we haven’t got comprehensive results due to it was our preliminary exploration. Here, we presented the pigments concentration changes in WT and HY5-OE background when silencing GSNOR (figure a&b), and the expression of PORC and PSY2 in SNP treated HY5-OE plants (figure c). Please see the attachement.

Point 4: While quantifying the fluorescence values with DAF-FM-DA staining (Figure 1E and F), whether the authors calibrated the autofluorescence due to chlorophyll? In figure 1F, the “horizontal scale bar” is missing.

Response 4: We are grateful to you for pointing out this problem. The autofluorescence of chloroplasts was captured at 585 nm in previous research (Floryszak-Wieczorek, J et al, 10.1007/s00425-006-0321-1). The emission wavelength we used was 525nm to avoid the autofluorescence of chloroplasts to the greatest extend. Nevertheless, we did a negative control without the NO-specific probe. Results showed that the autofluorescence of chloroplasts did exist, but was weaker than the fluorescence with DAF-FM-DA staining. The positive fluorescence could be found in cells, not only in the chloroplasts (figure d). Microscope, laser and photomultiplier settings were held constant during the course of the experiment in order to obtain comparable data. In addition, the scale bar has been added in figure 1C in the revised MS (figure 1F in the original version). Please see the attachement.

Point 5: The expression level of the PORC and PSY2 in the hy5 background is around 50~60% of WT. From this observation, it can be speculated that there are other factors which maintain a stable expression of PORC and PSY2 during pigment biosynthesis. So, in that case, HY5 is not indispensable in the regulation of pigments biosynthesis by NO. At this stage, it is too rudimentary to conclude that HY5 is the only candidate downstream of NO pathway.

Response 5: We agree with you that HY5 is not the only candidate downstream of NO pathway. Combining the responses to major points 1, 2, and 3, we considered that there are other potential factors downstream of NO to regulate pigments biosynthesis, while HY5 still plays the dominant role. In addition, the novelty of this work is the role of high NO concentrations in the HY5 stability, and subsequent decrease of photosynthetic pigments and efficiency. Therefore, we focused on the relationship between NO and HY5 in our research. To eliminate the potential confusion, as pointed by the reviewer, we have rewritten the MS and have added some information in Discussion, avoiding some one-sided statements.

Minor:

Point 6: I wonder whether the authors included the possibility of COP1 in degradation of HY5 in this NO-HY5 mediated pathway that fine-tunes pigment biosynthesis.

Response 6: Many thanks for your concern. We have taken this possibility in to account. We observed the influence of NO donors on the subcellular localization of GFP-COP1 in tobacco leaves both in dark or light. Results showed that SNP and GSNO improves COP1 nucleus-enriched localisation in the light (figure e). We will combine your suggestion to conduct in-depth research in our following works.

Point 7: English correction and typos need to be corrected: For example, in line #31, “such biosynthesis in cellular always occurs”, I guess it would be cellular organisms.

Response 7: Thank you for your suggestion. We have modified the sentence in the revised MS.

Point 8: Line #36, light-mediated seedling greening. The “d” is missing from mediated.

Response 8: We have corrected the mistake.

Point 9: In figure S2B, what are MG, BK, BK+3, BK+7? Please mention it in the legend.

Response 9: We are sorry for the mistake and have added the meanings of MG, BK, BK+3, and BK+7 in the figure legends of S2B.

Point 10: In figure 5, the legend “A” is missing.

Response 10: We are sorry for the mistake and have corrected it in the revised MS.

Point 11: In figure 5, please mention which panel is H2O, SNP and GSNO treatment. Saying them in the original image file in figure 5B is not enough.

Response 11: We are sorry for the mistake and have corrected it in figure 5B.

You can find the figures in this response in the attachment. 

Reviewer 2 Report

The work of Wang et al, describe the effect of high nitric oxide (NO) concentrations over HY5 stability and photosynthetic pigments in tomato. 

The manuscript is well written, the English is understandable, with only few details to review. It is well performed, the logic of the experiments is suitable, Figures are well performed, the results are correctly interpreted, and conclusions are consistent with these interpretations. The bibliography is adequate and actualized,

In general, this manuscript will be interesting to the readers in the field.

However, in my opinion, some of the results presented here may be considered incremental, and the manuscript must avoid to repeat information. By example:

-In the Results section, line 119 to 182, and Figures 1 and 2.

Item was entitled: . 2.1. Plant photosynthetic growth is inhibited by both endogenous and exogenous NO in tomato

 However, results show and discuss the effect of endogenous and exogenous NO mainly as differences in chlorophyll and carotenoid content, as well as in PSY and POC expression.  There is a reference to photosynthetic efficiency only  in lines 135 to 139. As cited in the manuscript, NO increasing and photosynthetic efficiency decay by GSNOR knock out were already reported in Arabidopsis (Chen et al, 10.1038/cr.2009.117, Shi et al, 10.1016/j.molp.2015.04.008) and tomato (see figs 1 and 9 of ref. 19 in this manuscript). Although GSNOR knock out was done by a different method in the present work, some results are identical.

I believe that title in line 119 could be changed, and the entire paragraph (lines 119 to 182)must be re written focusing in the novelty, avoiding reported results as Fig 1A,D,E,F.

-Item 2.3 lines 211 to 245 and figure 4,

2.3. HY5 is a transcriptional activator of PORC and PSY2

Again, I believe that most of the information in item 2.3 could be incremental. The role of HY5 as regulator of carotenoid and chlorophyll, as well as the binding of HY5 to PORC, PSY and other promotors was deeply shown in Arabidopsis (see figs 1-4, 7 and 8 in ref 34). Why must be different in tomato? Moreover, some of the deleterious effects of HY5 deletion have already been reported by the authors (reference 52 in this manuscript).

Additionally, in lines 352-366 of Discussion, the authors say:

“The data presented here show that HY5 is the key factor regulates chlorophyll and carotenoid accumulation in tomato leaves.” (Lines 355-356)”

I am not able to see the novelty.  I think that the item (lines 211 to 245) should be deleted, and figure 4 incorporated as supplementary information.

In my opinion, the novelty of this work is the role of HIGH NO concentrations in the HY5 stability, and subsequent decrease of photosynthetic pigments and efficiency.The results show that HY5 effect could be suppressed by a NO-driven HY5 degradation. I agree with the authors that:

“The finding that NO regulates the expression of genes involved in photosynthetic pigments biosynthesis is a breakthrough in this research.” (lines 349-350)

However, NO was increased to abnormal high concentrations either by adding 500 µM of SNP or GSNO (Figure 2A-C), or by GSNOR mutation generated by a CRISPR/Cas9 technique. Moreover, GSNOR mutants are lacking of the main NO detoxifying mechanism. and loss of GSNOR function leads to compromised phytohormone signalling, including auxin, ethylene and abscisic acid (see Table 1 in ref 19, and references therein).

So, my question is: which is the physiological relevance of these results? Is there any physiological condition where NO level could be highly increased in the absence of a detoxifying mechanism? It will be very interesting if authors could consider this argument in the discussion.

Minor details

1)Title could be changed to

High nitric oxide concentration inhibits photosynthetic growth by promoting the
degradation of transcription factor HY5 in tomato,
or something similar, taking in account that HY5 degradation  is possible only at high NO concentrations.

2) In Figure 2, NO levels were detected 5 days after treatments with SNP, GSNO or water. SNP and GSNO are transient effectors and maximal NO liberation is reported 1 day after treatment (Floryszak et al, DOI 10.1007/s00425-006-0321-1). NO measured will be coming from donors or may be a consequence of an initial NO burst?

3) Line 180-181: “All these biochemical observations along with genetic evidence confirm that GSNOR modulates NO accumulation in cellular”. Line 329-330: “The gsnor-knockdown mutants used in this study were disabled to scavenge superfluous NO in cellular and kept a high level of NO”.  What does “cellular” mean?

4) Figure 3 F: How long after SNP or GSNO treatment was HY5 analyzed?

5)Figure 5: Results in panel A and B are the same. The right figure is in original images. Replace panel B. Moreover, MG132 effect in cell-free extracts could not be representative of in vivo inhibition. Could be gsnor treated with the inhibitor before HY5 immunoblot  (like figure 3E)?

6)Figure 8: GONOR must be replaced by GSNOR.

Author Response

Response to Reviewer 2 Comments

Thank you for your valuable suggestions and comments, which assisted us in improving our MS. We have added required modifications to the manuscript in order to accommodate all the suggestions for improvement. The changes to the revised manuscript are listed below:

Major:

Point 1: In the Results section, line 119 to 182, and Figures 1 and 2. 

Item was entitled: 2.1. Plant photosynthetic growth is inhibited by both endogenous and exogenous NO in tomato However, results show and discuss the effect of endogenous and exogenous NO mainly as differences in chlorophyll and carotenoid content, as well as in PSY and POC expression. There is a reference to photosynthetic efficiency only in lines 135 to 139. As cited in the manuscript, NO increasing and photosynthetic efficiency decay by GSNOR knock out were already reported in Arabidopsis (Chen et al, 10.1038/cr.2009.117, Shi et al, 10.1016/j.molp.2015.04.008) and tomato (see figs 1 and 9 of ref. 19 in this manuscript). Although GSNOR knock out was done by a different method in the present work, some results are identical. I believe that title in line 119 could be changed, and the entire paragraph (lines 119 to 182)must be re written focusing in the novelty, avoiding reported results as Fig 1A,D,E,F. 

Response 1: According to your suggestion, we have changed the title of item 2.1 into “2.1. Photosynthetic pigments biosynthesis is inhibited by both endogenous and exogenous NO in tomato”. To better focus on the novelty, we rewritten the whole paragraph and changed the order of figures in Figure 1 in the revised article.

Point 2: Item 2.3 lines 211 to 245 and figure 4,

2.3. HY5 is a transcriptional activator of PORC and PSY2

Again, I believe that most of the information in item 2.3 could be incremental. The role of HY5 as regulator of carotenoid and chlorophyll, as well as the binding of HY5 to PORC, PSY and other promotors was deeply shown in Arabidopsis (see figs 1-4, 7 and 8 in ref 34). Why must be different in tomato? Moreover, some of the deleterious effects of HY5 deletion have already been reported by the authors (reference 52 in this manuscript).

Additionally, in lines 352-366 of Discussion, the authors say:“The data presented here show that HY5 is the key factor regulates chlorophyll and carotenoid accumulation in tomato leaves.” (Lines 355-356) I am not able to see the novelty. I think that the item (lines 211 to 245) should be deleted, and figure 4 incorporated as supplementary information.

Response 2: According to your suggestion, we deleted the item 2.3 and added figure 4 into supplementary information figure S6. In order to make the structure of our MS clearer, we made some modifications by changing the initial item 2.4 into item 2.4 and 2.5 in the revision. In addition, researches of the carotenoid pathway in tomato focused on PSY1, which highly expressed in fruit ripening progress (Llorente, B et al, 10.1111/tpj.13094). It was also been reported that the green tissue-specific SlPSY2 is more effective for photosynthesis (Cao et al, 10.1104/pp.19.00384). As a result, PSY2 was the target gene during tomato vegetative growth in our research.

Point 3: In my opinion, the novelty of this work is the role of HIGH NO concentrations in the HY5 stability, and subsequent decrease of photosynthetic pigments and efficiency.The results show that HY5 effect could be suppressed by a NO-driven HY5 degradation. I agree with the authors that:

“The finding that NO regulates the expression of genes involved in photosynthetic pigments biosynthesis is a breakthrough in this research.” (lines 349-350)

However, NO was increased to abnormal high concentrations either by adding 500 µM of SNP or GSNO (Figure 2A-C), or by GSNOR mutation generated by a CRISPR/Cas9 technique. Moreover, GSNOR mutants are lacking of the main NO detoxifying mechanism. and loss of GSNOR function leads to compromised phytohormone signalling, including auxin, ethylene and abscisic acid (see Table 1 in ref 19, and references therein).

So, my question is: which is the physiological relevance of these results? Is there any physiological condition where NO level could be highly increased in the absence of a detoxifying mechanism? It will be very interesting if authors could consider this argument in the discussion.

Response 3: In response to your valuable suggestion, we have added some information in Discussion as follows: Additionally, phytohormone signaling might be involved in GSNOR-mediated plant growth and development. The ethylene productivity rate was increased in GSNOR-RNAi plants [19]. It was reported that ethylene facilitated NO generation by activation of both NR and NOS-like in Arabidopsis and Tagetes erecta L.[40, 41]. The crosstalk between NO and other phytohormones might be a potential mechanism that maintains the high NO level in gsnor knockdown plants.

Minor details:

Point 1: Title could be changed to:

High nitric oxide concentration inhibits photosynthetic growth by promoting the degradation of transcription factor HY5 in tomato, or something similar, taking in account that HY5 degradation  is possible only at high NO concentrations.

Response 1: We are appreciated with your suggestion and we have changed the title into “High nitric oxide concentration inhibits photosynthetic pigments biosynthesis by promoting the degradation of transcription factor HY5 in tomato”.

Point 2: In Figure 2, NO levels were detected 5 days after treatments with SNP, GSNO or water. SNP and GSNO are transient effectors and maximal NO liberation is reported 1 day after treatment (Floryszak et al, DOI 10.1007/s00425-006-0321-1). NO measured will be coming from donors or may be a consequence of an initial NO burst?

Response 2: Thanks for your concern. Surely, as you said, SNP and GSNO are transient effectors. Considering the pigments content were measured at the fifth day, we chose the indicated time in our MS. To maintain the NO level, SNP and GSNO was sprayed once every other day during the treatment. We are sorry for the omission of description in Pharmacological treatments and have added this information in the revised MS.

Point 3: Line 180-181: “All these biochemical observations along with genetic evidence confirm that GSNOR modulates NO accumulation in cellular”. Line 329-330: “The gsnor-knockdown mutants used in this study were disabled to scavenge superfluous NO in cellular and kept a high level of NO”.  What does “cellular” mean?

Response 3: We are sorry for the unclear description. We have changed “NO in cellular” into “intracellular NO” in the revised MS.

Point 4: Figure 3 F: How long after SNP or GSNO treatment was HY5 analyzed?

Response 4: Both the mRNA and protein level oh HY5 was analyzed 3 hours after the treatment. The information has been added in the legend of figure 3 in the revised MS.

Point 5: Figure 5: Results in panel A and B are the same. The right figure is in original images. Replace panel B. Moreover, MG132 effect in cell-free extracts could not be representative of in vivo inhibition. Could be gsnor treated with the inhibitor before HY5 immunoblot (like figure 3E)?

Response 5: Thank you for making this critical point. As your suggestion, we treated the WT and gsnor mutants with cycloheximide (CHX). The result of HY5 immunoblot showed that the initial HY5 protein content in gsnor was too little and is hard to detect the changes during the treatment. Immunoblot analysis of endogenous HY5 protein (figure 3E,G) could also prove the in vivo inhibition of HY5 by NO. In addition, the mistake in figure 5B has been corrected in the revised MS.

Point 6: Figure 8: GONOR must be replaced by GSNOR.

Response 6: We are sorry for the mistake and have corrected it in Figure 8 (Figure 7 in revised MS).

Round 2

Reviewer 1 Report

Thank you for addressing the concerns. The manuscript is substantially revised in the present condition. 

Hopefully, this work can give a new insight into the NO-HY5 pathway involved in photosynthetic pigment biosynthesis. 

Reviewer 2 Report

The authors have responded satisfactorily to my comments, and I hope that they agree with me that this is an improved version of the manuscript.

Please check  the englih of this new version. By example, in line 136-138:

"We deter- 136
mined the contents of chlorophyll and carotenoid, two important photosynthetic appa- 137
ratus."

That is no clear.